# Finding My Drumbeat: Applying Lessons Learned from Remo Ruffini to Understanding Astrophysical Transients

Chris Fryer 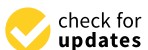

Center for Theoretical Astrophysics, Los Alamos National Laboratory, Los Alamos, NM 87545, USA;
fryer@lanl.gov; Tel.: +1-505-500-2279

**Abstract:** As with many fields from fashion to politics, science is susceptible to "bandwagon"-driven research where an idea becomes increasingly popular, garnering a growing amount of "scientific" support. Bandwagons allow scientists to converge on a solution, but when the prevailing bandwagon is incorrect or too simple, this rigid mentality makes it very difficult for scientists to find the right track. True scientific innovation often occurs through scientists willing to march to the beat of their own drum. Using examples in the field of astrophysical transients, this paper demonstrates the importance of supporting scientists in their quest to develop their own personal drumbeat.

**Keywords:** gamma-ray bursts; supernovae; neutron stars; black holes

## 1. Introduction: Bandwagon Science

The bandwagon effect is a psychological phenomenon in which people perform something primarily because other people are performing it, regardless of their own personal beliefs. This tendency of the alignment of beliefs or viewpoints is also referred to as herd mentality. Although the bandwagon effect was originally used to describe political viewpoints, it occurs in a broad range of situations, including scientific research and funding.

Bandwagon approaches in science can be beneficial to a scientific endeavor. For instance, when new technology or new methods become available (e.g., machine learning, quantum computing, etc.), many funding agencies jump on a bandwagon to fund this research. Funding managers place much of their funds into calls that require this bandwagon science, and scientists who embrace bandwagon science tend to only review highly the science that is on that bandwagon. The strength of this approach is that it directs funding to an up-and-coming topic, jumpstarting its research. The problem with this is that oftentimes this influx of funding is far too great to drive any productive science, and this overfunding one field leaves other fields unfunded. For many funding agencies, the "bandwagon-of-the-month" oscillates rapidly enough that building and retaining scientists to work on the projects is impossible and much of the funding is wasted.

Science fields can also follow bandwagons. These bandwagons generally support a "standard paradigm" explaining some phenomena. The advantage of the bandwagon effect for such standard paradigms is that it focuses research on a specific paradigm. In principle, this research focus will more quickly flesh out the details of a current standard-paradigm. The disadvantage is that this focused research often poorly identifies weaknesses in the paradigm. Because of this, standard-paradigms often remain "standard" long after the evidence clearly shows it is incorrect. In addition, bandwagon science tends to oversimplify models. Typically, progress in these fields occurs because a subset of scientists do not join the bandwagon, preferring to march to the beat of a different drum. In the end, science progresses through a cacophony of drumbeats and, as messy as this approach is, it may be a requirement in the evolution of science.

In this paper, we review a series of examples in astrophysical transients where science progress required scientists willing to move against the bandwagon. Remo Ruffini's approach to science epitomizes this alternative drumbeat and Section 2 discusses how this approach has driven research in gamma-ray bursts (GRBs). His example is an inspiration driving advances and discoveries of supernovae (SNe): Section 3. This paper focuses on these two examples, but bandwagons exist throughout science. We conclude with a broader discussion of the implications for science as a whole.

## 2. Gamma-Ray Bursts

The field of GRBs has been prone to bandwagon beliefs that often oversimplify the physics. Although these simplifications make it easier to explain a model, they often neglect physics or processes that are crucial in interpreting observations, ultimately delaying progress in the field. In some cases, the standard-paradigm science case was much weaker than the actual science case without the bandwagon simplifications, making it more difficult for the GRB community to justify new instruments and science studies. As such, the history of GRB science provides examples of how a paradigm can dominate a field and the importance of science outside of the standard paradigm.

### 2.1. Gamma-Ray Emission

GRB emission models and the reluctance of that community to envision solutions beyond the standard mechanism is a prime example of how bandwagon science can damage a field. Figure 1 shows the basic picture of a GRB outflow. In the standard model, synchrotron (and synchrotron self-Compton) emission produces both the prompt and afterglow emission [1,2]. In the prompt phase, this model argues that the chaotic emission is produced by variability in the jet power, presumably caused by instabilities in the accretion disk. The variability produces outbursts of different Lorentz factors. When a faster burst hits a slow burst, particle acceleration in the shock produces nonthermal emission that is Lorentz-boosted into the gamma-rays. In this standard paradigm, the afterglow (post-burst of gamma-rays) emission is caused by synchrotron emission as the shock propagates through circumstellar medium.

This model has a lot of strengths. First, with a basic synchrotron (or synchrotron plus self-Compton) model incorporating a range of simply-described parameters, the data can be fit fairly well [2]. In its simplest form, this model could be used to probe the jet Lorentz factor from the prompt emission, and, within this paradigm, the afterglow emission can be used to probe the circumstellar medium; however, this model has a number of issues. The parameters needed to explain the GRB prompt emission (Lorentz factor and fraction of power-law electrons) pushed the limits of what can be produced by both engine and plasma kinetics models. The predictions from the afterglow models argued for circumstellar density profiles that contradicted the strong wind models from the standard collapsar progenitor [3]. For the most part, given these difficult-to-explain results, many of the GRB engine and progenitor theorists moved on to other fields, and progress in our understanding of GRBs stalled. Incredible results require powerful arguments and the simplified bandwagon science did not provide these arguments.

Alternative models exist. At early times, some teams argued that external shocks can explain the prompt emission [4–6]. In these models, variability relied on inhomogeneities in the circumstellar medium. Although seemingly ad hoc, these inhomogeneities are what has been predicted by stellar models [7–13]. Because GRB progenitors are believed to have stronger outflows than normal SNe, these inhomogeneities should be most extreme in long-duration GRB progenitors. These features did not fit into the standard paradigm and, often when something does not fit in a paradigm, features are thrown out with the noise. Thus, even though this signal should be stronger in GRBs than SNe, the first evidence of these inhomogeneities came from early-time emission and shock breakout in supernovae [14–16]. Strict adherence to the GRB standard model prevented GRB observers

from noticing evidence of this important aspect of GRB progenitors. The science carried out with GRBs was limited by the dominance of GRB bandwagon scientists.

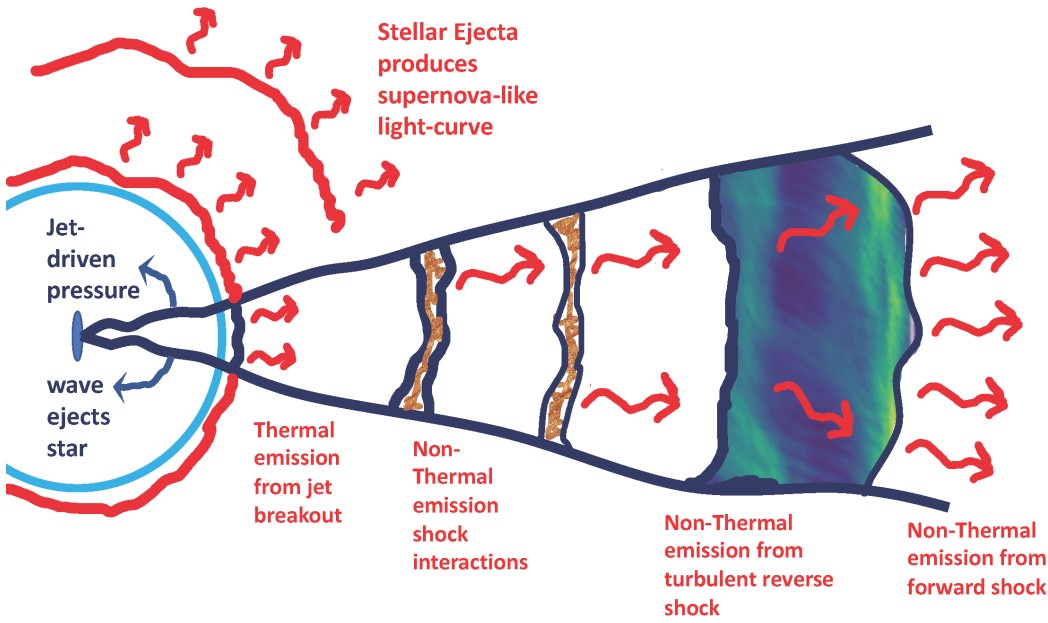

**Figure 1.** Diagram summarizing many of the emission mechanisms in a GRB-event from a massive star. The standard paradigm assumes that nonthermal emission caused by shock interactions of different outbursts produced in an unstable accretion-driven engine produces the prompt emission and the shock interactions of the jet with the surrounding medium produce the afterglow. However, other sources exist, e.g., nonthermal emission from turbulent particle acceleration in the reverse shock, and thermal emission from the breakout of the shock from the star. The supernova-like explosion that occurs with these explosions occurs because of both the fact that a pressure wave wraps around the star and the disk drives a wind, ejecting the stellar material. Although this is not a standard supernova engine, this nonrelativistic ejecta produces supernova-like or hypernova emission.

It is also possible that turbulence in the reverse shock can cause the needed variability, but this has not been studied in detail. In SNe, this turbulent reverse shock region is also a site of particle acceleration: cosmic ray and high-energy neutrino production [17]. If we could disentangle this portion of the emission from the rest, we might be able to better understand the production of cosmic rays and high-energy neutrinos in GRBs.

Another model that received very little attention was the role of thermal emission in gamma-ray bursts. As we have learned with neutron star (NS) mergers, the emission arises from a range of sources including thermal (powered by shock heating, radioactive decay, and possibly additional engines such as magnetars) and nonthermal (both from the jet as discussed above and the remnant) emission. Initially, the GRB community ignored studies suggesting that the emission (or parts of the emission) could be powered by thermal emission [18–20]. In part, because the theory behind the thermal emission is much better understood than the nonthermal emission, it is a powerful probe of the properties of the GRB ejecta. For example, Figure 2 shows the spectra from this ejecta, assuming a distribution of Lorentz factors where the area (as a function of Lorentz factor) of the thermal emission is $\propto \Gamma^{\alpha}$. Here, we assume that the minimal Lorentz factor is 1 and the maximum Lorentz factor is either 100 or 200 (for more details, see Lesage et al. in preparation). By studying the emission spectra, we can constrain the structure of the GRB jet at a level that has not been carried out to date, but this requires disentangling the thermal and nonthermal components. This work was delayed until scientists rediscovered the concept of a thermal component that produces the gamma-ray signal [21,22]. It is now applied to many of the new GRB observations, e.g., [23,24].

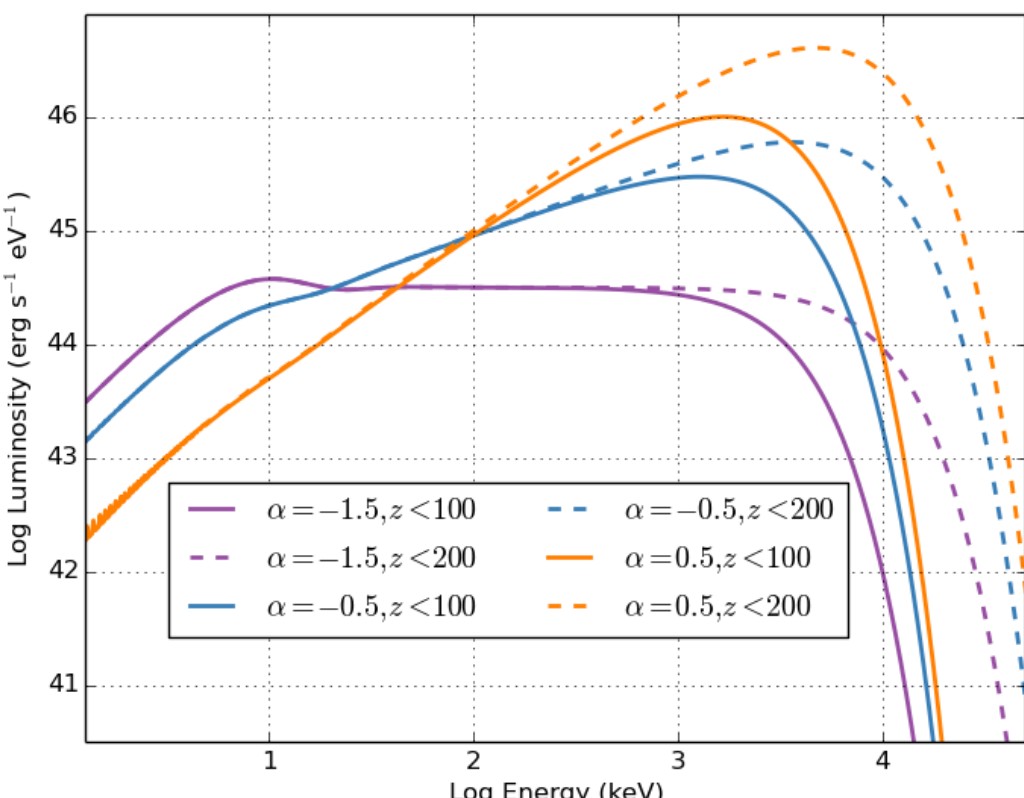

**Figure 2.** Lorentz-boosted spectra from thermal emission assuming different distributions of the Lorentz factors in the ejecta. The distributions are assume to follow a power law: $\propto \Gamma^{\alpha}$.

Figure 1 also shows emission from the off-jet axis ejecta. The jet sends a pressure wave through the star and this alone has been shown to be able to eject the stellar material [25], but the accretion disk is also likely to blow a strong wind that contributes to the ejection of the stellar material. This mass ejection both powers a supernova-like light curve and, ultimately, limits the accretion, cutting off the engine. Note that a supernova is not required to produce this emission, just the ejection of the stellar material and the pressure wave produced by the jet is sufficient to achieve this.

In this case of GRB observations, by not considering different components to the emission of GRBs, bandwagon scientists delayed progress in our understanding of this emission and limited what we could learn from the data. Many theorists moved on to other fields, believing that nothing new could be learned from the emission itself. In turn, this weakened the science case for further studies, limiting future progress in this field. Pioneers in alternative models have started to revive this field, demonstrating the wealth of information we can gain from the prompt and afterglow emission if we are willing to accept the complex picture of emission mechanisms.

### 2.2. GRB Engines

The engine or power source for gamma-ray bursts has, for the most part, avoided the trend of bandwagon science to restrict the models studied. A number of broad engine scenarios have been proposed, but two basic engines have been the primary focus in the astrophysical literature: black hole accretion disks (BHADs) and magnetars. The biggest issue with bandwagon beliefs in GRB engines is that scientists have too-easily accepted these models without bearing in mind the properties of each of these models. The refinement of models used the engine properties and weighed the strengths and weaknesses of each model with respect to available data. As the data, and our physical understanding of the engine, evolve, we must reassess each model.

Let us review some of the model properties to better understand this concept (summary in Table 1). Black hole accretion disk models were initially highlighted because they matched (and still match) many of the observed properties of GRBs and it was easier to identify progenitors for these models satisfying the angular momentum requirements (for a review, see [26]). It can produce the observed power (with sufficient beaming) and, because the jet can avoid the accretion disk wind, the model can avoid baryonic contamination and produce high Lorentz-factor flows. The power of this engine depends upon the energy in the disk. Once the mass reservoir in the disk is depleted (no more accretion), the energy in the disk disappears and the engine turns off (limiting the GRB duration). The energy in the disk also decreases with the radius (and hence mass) of the black hole. Collapsar and other massive star disks are fueled by the continued accretion supply from the infalling star, producing long bursts. The compact disks from neutron star mergers accrete quickly and the engine will turn off quickly, producing short bursts only. This simple picture helps explain the different burst-duration populations, but now the data appear to be suggesting a different picture. For example, some GRBs that show evidence of a neuron star merger are occurring in long bursts [27–29].

**Table 1.** Engine weaknesses and strengths.

| Engine | Power | Lorentz Factor | Duration | Formation |
|--------|-------|----------------|----------|-----------|
| BHAD | Yes (depending on beaming) | Yes (disk wind is off-axis) | Duration limited to accretion timescale and black hole size | |
| Magnetar | Yes (depending on beaming and rotation) | ??? (must overcome neutron-star wind) | Duration can extend beyond accretion | Needs high rotation rates; Why does accretion not bury magnetic fields? |
| NSAD | Yes (simiar to BHAD) | ??? (must overcome neutron-star wind) | Duration limited to accretion timescale (NS collapse features?) | Needs high rotation rates |

The magnetar engine's strength lies in its ability to explain this long-lived emission (or plateau phase) in neutron-star merger GRBs, but this engine has a number of difficulties. It typically invokes the same progenitor scenarios where accretion is occurring at the same time as the engine is produced, but we know from pulsar studies that this accretion will bury the magnetic fields [30]. The hot neutron star in this scenario also drives a wind that the magnetic engine must plow through, and this propagation is likely to sweep up mass, lowering the Lorentz factor of the jet [31]. Finally, obtaining the power requires very high angular momenta which, although easily achievable in neutron star merger progenitors, is harder to achieve with massive-star scenarios.

Neutron star accretion disk (NSAD) engines are very similar to BHAD engines. The energy in the disk around a neutron star is very comparable to a $3\,M_\odot$ black hole, and the energetics are comparable. However, because there is more angular momentum in the outer layers of a massive star, it is more difficult to produce disks for this engine. This model also has the same baryonic contamination problem that the magnetar engine has.

Other engines exist, invoking pair-plasma power source from an induced gravitational collapse [32–35]. Although quite a few papers have studied this engine, this work was limited to a single team. The potential of this engine mandates further study.

Many times, bandwagon science leads to scientists forgetting the reasons for the standard paradigm, causing scientists to hold on to a simple model for too long and making it difficult to determine key observational tests of the models and distinguish between standard models and newly proposed models.

*2.3. GRB Progenitors*

As we have seen in the GRB emission picture, bandwagon science focuses on a simple picture even if our fundamental physics understanding argues the problem is much more complex. This trend of bandwagon physics is clearly evident in studies of GRB progenitors of the BHAD engine. Although a broad range of BHAD progenitors were discussed in [36], the community focused on just two scenarios: neutron star mergers for short-duration bursts and collapsars for long-duration bursts. It was argued that the high accretion rates but low accretion timescales for neutron star mergers better fit short–hard bursts and the lower accretion rates with longer disk-feeding timescales of collapsars matched the long–soft bursts [37]. This two-component BHAD paradigm further predicted that long bursts would occur in star-forming regions and short bursts would occur out of these regions and even offset from the host galaxy [36,38].

We already discussed the issues with the BHAD scenario in explaining the long-lived emission (plateau phase) of short bursts, but the simplified collapsar-only progenitor scenario also has difficulties explaining long bursts. First, it remains unclear how the cores of massive stars can collapse with enough angular momentum to form a disk [1]. In addition, it remains true that the supernova-like outbursts accompanying long-duration gamma-ray bursts tend to lack hydrogen and helium features (Type Ic supernovae). The fact that the star had to be a compact (Wolf–Rayet) star was part of the collapsar model. Since the jet in the massive star model must clear out a baryon-free channel, the engine must be able to last at least as long as this clearing phase. Even if a jet is driven in a hydrogen giant star, it will not break out of the star and clear out a low-baryon jet region before the engine turns off. As such, it will not produce large amounts of gamma-ray emission. This is problematic for a giant star, but not for compact helium or carbon/oxygen stars [39]. However, the fact that observations of SNe associated with GRBs are only type Ic SNe argues that only carbon/oxygen stars form GRBs. In the standard collapsar paradigm where the mass-loss occurs through winds, it is difficult to explain why helium stars (which produce type Ib SNe) could not also form GRBs. Wind mass-loss, which tends to remove angular momentum, made it even increasingly difficult to produce black hole accretion disks. Ultra-long bursts are extremely difficult to explain with this engine.

Despite these striking deficiencies, many studies in the field ignore alternative progenitor scenarios. Table 2 shows a subset of the progenitors studied in [36]. Most of the massive star models struggle to explain why the long bursts are mostly associated with supernovae with no helium lines. The exception may be the common envelope formation scenario invoking tidal locking to spin up the core. In this case, sufficient angular momentum may only occur if an extremely tight binary (one consisting of a carbon/oxygen star, a so-called ultra-compact star) is produced. In this scenario, binary interactions both eject mass and spin up the massive star so that it will form a disk when it collapses to a black hole. The common envelope scenario argues that when the massive star in a massive star/compact remnant (NS or BH) binary expands to a giant phase, it envelopes the compact remnant. The subsequent orbital inspiral ejects the envelope (using orbital energy to drive the mass ejection). This can remove the hydrogen envelope and produce a tight binary. If we want to produce a carbon/oxygen star, we must undergo a second common envelope, assuming the helium star also expands into a "giant-like" phase, enveloping the compact remnant. The subsequent common envelope phase would produce an even tighter binary where tidal locking could rapidly spin up the carbon/oxygen star. The problem is that, at this time, only low-mass helium stars undergo giant phases. These low-mass stars are not expected to collapse to form black holes. It might be that this scenario only works for NSAD engines.

**Table 2.** BHAD progenitors.

| Scenario | Duration | Location | Angular Momentum | Associated Transient | Circumstellar Medium |
|---|---|---|---|---|---|
| Massive Star | | | | | |
| Wind Mass-Loss | Long bursts | Star-forming regions | Difficult/impossible | Type Ib/Ic | Wind profile |
| Common Envelope | Long bursts | Star-forming regions | Tidal spin-up | Type Ib/Ic (Tidal Spins could limit to Ic) | Wind plus shell |
| He–He Merger | Long bursts | Star-forming regions | Difficult | Type Ib/Ic | Wind plus shell |
| Helium Merger | Long and ultra-long bursts | Star-forming regions and slightly beyond | Can have too much angular momentum | Type Ib/Ic | Wind plus shell |
| Binaries | | | | | |
| NS/NS | Short bursts | Off-set | $\sim$10 km disk | Disk ejecta only (Kilonova) | Interstellar or intergalactic medium |
| NS/BH | Short bursts | Off-set | Disk forms for subset | Disk ejecta only (Kilonova) | Interstellar or intergalactic medium |
| WD/(BH/NS) | Long bursts | Mild off-set | $\sim$10,000 km disk | Fast supernova from disk wind | Mostly interstellar medium |

The only massive star progenitor that consistently produces sufficient angular momentum is the helium merger model, where a compact remnant inspirals within a star but merges without ejecting all of the envelope. As it spirals into the center of the massive star, it behaves very similarly to a collapsar accretion scenario. Because of this similarity, this He-merger model was conflated with the collapsar model.

At this time, only the binary-driven hypernova progenitor for the pair-plasma engine has a natural explanation for the hydrogen and helium-poor supernovae associated with GRBs [40]. This alone argues that this engine warrants further study.

Why is it so important to study the exact details of the progenitor instead of relying on the simple bandwagon scenario? First, without identifying the features of each progenitor and its ability to explain GRBs, funding for this research has faltered and little progress has been made in better understanding GRBs, and, in addition, because the different progenitor scenarios have different dependencies with redshift. Many studies assume a given progenitor, e.g., [41] and many of these do not even understand that they have chosen a specific progenitor. Because of this oversimplification in the bandwagon, many of the studies and expectations of metallicity and redshift evolution of GRBs are flawed.

## 3. Supernovae

GRBs are not the only field in astronomy where a bandwagon focus on simplifying models has led to both misinterpretations of the data and a delay in scientific progress. Studies of supernovae have experienced similar problems, but for many of us who have worked in both fields, lessons learned from the GRB community have helped us overcome issues in the supernova field.

### 3.1. Thermonuclear Supernovae

The field of thermonuclear supernovae provides a classic example of where bandwagon science not only oversimplified the physics but pushed a paradigm that ultimately is now believed to be just one solution to the problem. For thermonuclear supernovae, the

power source is nuclear burning (the conversion of the carbon/oxygen of a white dwarf to silicon and nickel). How this material ignites and burns is not known, but many ideas exist: many-ignition-site deflagration, deflagration transitioning to detonation, detonation of an accretion layer driving the compression of the carbon/oxygen core (e.g., sub-Chandrasekhar models), deflagration in the core igniting a detonation in the accretion layer that then drives a detonation of the core (e.g., gravitationally confined detonation), collisions, double degenerate mergers, etc. (see, for example [42,43]). Although all engines rely on a carbon/oxygen white dwarf, the properties of the white dwarf (e.g., mass), the engine, and the progenitors all vary considerably.

This broad set of models suggests a vibrant field unconstrained by bandwagon science, but this is only after a period of extreme constraints caused by bandwagon science. In the early 1990s, the best light-curve models compared to data strongly supported a Chandrasekhar-mass progenitor for thermonculear supernovae [44,45]. This led the field to primarily consider only engines and progenitors within this paradigm for nearly 10–15 years. Indeed, breaking out of this bandwagon required both the abundance of evidence pushing away from the standard paradigm (including progenitor studies) and a strong-willed push by teams pushing alternative models, e.g., the FLASH team [46].

### 3.2. Core-Collapse Supernovae

Although the core-collapse supernova (CCSN) community has also homed in on a single paradigm where convection above a collapsed core enhances the conversion of the gravitational potential energy released in the collapse of the core of a massive star into kinetic energy of the explosion [47], this field seems to better allow alternative explosion mechanisms, including jet engines [48]. Indeed, the lack of progress in alternative explosion mechanisms has been more driven by (a) the difficulty in conducting magnetohydrodynamic models and (b) evidence from observations that most stars do not have enough angular momentum for these alternative mechanisms to explain most supernova observations (although such engines probably explain a subset of the observations). This is less an example of bandwagon restrictions and more of scientific limitations. The bigger issues with CCSN lie in interpreting the observations of both shock breakout and later-time light curves.

Shock breakout is the term used to describe the emission produced when the supernova shock produced in the central engine breaks out of a star. While in the star, the radiation is effectively trapped in the flow:

$$v_{\text{radiation}} = v_{\text{diffusion}} \approx \lambda c/D < v_{\text{shock}} \tag{1}$$

where the radiation velocity ($v_{\text{radiation}}$) is well described in the diffusion limit: $c$ is the speed of light, $\lambda$ is the mean free path, and $D$ is a fraction of the stellar radius. At the high densities of the exploding star, the velocity of the shock ($v_{\text{shock}}$) is much faster than this radiation velocity. As the shock breaks out of the star, the density decreases dramatically, increasing the mean free path and the effective radiation velocity. The radiation quickly changes from trapped in the flow to free-streaming out of the shock. The burst of light from this escape has been observed in a number of events and a simple analytic model was developed to infer the stellar radius [49].

This simple model was exciting for observers because a single observation of the duration and peak luminosity of shock breakout provides direct information about the progenitor star. The problem is that the simplification of the physics means that the the interpretations from the data are simply wrong. The shock breakout signal is affected by asymmetries in the shock [50], asymmetries in the star [51], and asymmetries in the stellar wind [15]. Observations of shock breakout [52] also show that the simple model for shock breakout is incorrect. Unfortunately, this means that the upcoming UltraSAT satellite will teach us less about stars than previously believed. UltraSAT data alone will not be able to disentangle this physics. Detailed models of shock breakout also show the limitations of proposed missions such as STAR-X to perform shock breakout science. In this case,

focusing too heavily on the simplified bandwagon model not only delays science, but leads to an inefficient use of scientific funding for new observatories.

Similarly, the light curves of supernovae also have suffered from too-simple models of the emission. Based on state-of-the-art models of light curves, supernova observers estimated the ejecta mass of core-collapse supernovae, arguing that the progenitor masses of these supernovae were all greater that $\sim$20 $M_\odot$ [53]. This directly contradicted models of the supernova engine [54]. In this case, theory pushed forward, ignoring these observed estimates. When direct observations of supernova progenitors [55] agreed with theoretical predictions, little time was lost through the simple paradigm model, but the transient community continues to overinterpret their data. For example, the transient community continues to assume that the peak luminosity of supernovae-like events in supernovae places constraints on the $^{56}$Ni yield (assuming that the decay of this radioactive isotope powers the light curve). However, other power sources exist, e.g., shock heating, and there is growing evidence that alternative energy sources must be understood to truly interpret supernova data. Figure 3 shows the luminosity of three supernova explosions, one powered by the decay of radioactive nickel and the other two using a simple shock deceleration model. In these simple models, the light-curve evolution appears very different but, especially for shock heating, a number of affects can alter the evolution. This affect has not been modeled in sufficient detail to determine its importance in SN light curves.

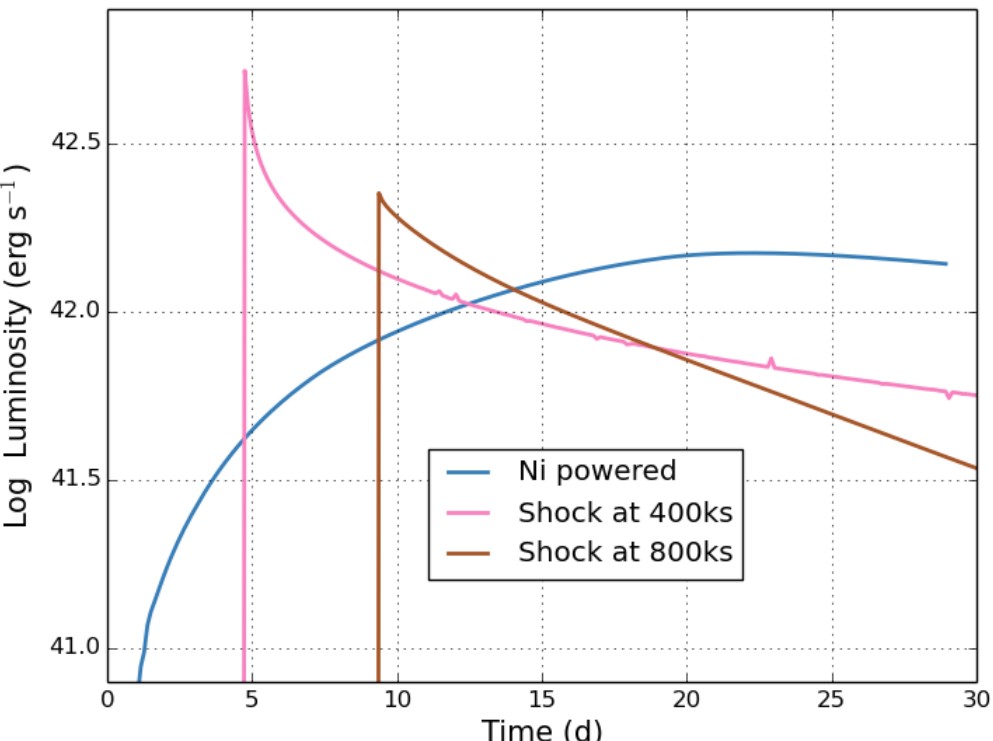

**Figure 3.** Luminosity for 3 SN explosions with the same Wolf–Rayet progenitor and explosion energy, but one is powered solely by the decay of radioactive nickel (0.05 $M_\odot$ of $^{56}$Ni) and the other two are powered by shock heating, where the shock heating model relies on a single shock decelerating the ejecta and converting kinetic energy into thermal energy that drives the emission. The shock heating models differ by having different timescales for the shock deceleration (400 and 800 ks). For more details, see Fryer et al. in preparation.

## 4. Conclusions

Science, similar to politics, is susceptible to bandwagon beliefs, where an often-simplified model becomes the standard paradigm and the bandwagon scientists are unwilling to consider deviations from this standard paradigm. We discussed examples in

astrophysical transients where bandwagon science has limited the science impact, delayed progress, and limited the development of the observations of a specific field, but these examples exist in all areas of research, including work by Gregor Mendel, Ignaz Semmelweis, Alfred Wegener, and George Zweig, to name a few.

Scientific funding is even more susceptible to bandwagon fallacies, often leading to inefficient use of precious research money. Funding managers tend to focus on bandwagon science ideas both because funding managers are enamored by these ideas and because they are easier to pitch to the governments that support their science. In addition, scientists who only conduct bandwagon science tend to be be overly critical of science and science proposals that are not part of existing bandwagons.

An example of such a funding trend may well be the current excitement over machine learning. Advanced statistical methods have their role in science. Funding managers are forcing scientists to apply one such method (machine learning) to their science by diverting their funds to research calls that require this research. Although it is likely that some new discoveries will come from this flux of funding, this overemphasis on a particular tool will also delay progress in many scientific fields.

Similar to Remo Ruffini, scientists must learn to "march to a beat of a different (their own) drum" and encourage others to also do so to ensure rapid scientific progress.

**Funding:** This work was supported by the US Department of Energy through the Los Alamos National Laboratory. Los Alamos National Laboratory is operated by Triad National Security, LLC, for the National Nuclear Security Administration of U.S. Department of Energy (Contract No. 89233218CNA000001).

**Data Availability Statement:** The data for the light-curve models shown here will be made available at the LANL Center for Theoretical Astrophysics transient emission site (https://ccsweb.lanl.gov/astro/transient/transients_astro.html, assessed on 5 March 2023) when the full database paper is published.

**Acknowledgments:** This paper honors Remo Ruffini scientists who have shown how "marching to their own drumbeat" can lead to innovation in science.

**Conflicts of Interest:** The author declares no conflict of interest.

## Abbreviations

The following abbreviations are used in this manuscript:

| | |
|------|------------------------------|
| NS   | Neutron Star                 |
| NSAD | Neutron Star Accretion Disk  |
| BHAD | Black Hole Accretion Disk    |
| GRB  | Gamma-Ray Burst              |
| BH   | Black Hole                   |
| SN   | Supernova                    |
| CCSN | Core-Collapse Supernova      |

## Note

[1] It is worth noting that magnetar and NSAD disk engines require even more angular momentum.

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
