# Peer review of "Finding My Drumbeat: Applying Lessons Learned from Remo Ruffini to Understanding Astrophysical Transients"

_universe, doi:10.3390/universe9060268_

Round 1
Reviewer 1 Report
Dear author! Thanks for the inspiring paper!
I have just a small general comment. Could you please express who are the main GRB theorists who left the field (eg. on lines 71 and following). This might help to find their original (and inspiring) papers.
I think (but it may be due to my poor english) that the sentence on lines 69-70 misses a "by. in the sense "what can be produced by both ...".
On line 106-107 it's written "is now being done" but the references are from 2011 and 2015.
On line 182, there's a mention of the plateu phase. Could you explain it better or expand this part?
There's a small typo to correct on line 182 "scneario”
Author Response
> I have just a small general comment. Could you please express who are the main GRB theorists who left the field (eg. on lines 71 and following). This might help to find their original (and inspiring) papers.
Authors I was thinking about were people like Stan Woosley or Thomas Janka, both of whom were very active in the late 1990s and early 2000s (counting their papers together, they published ~10 papers a year in this period), but both moved their focus to supernovae in the last 15 years. But I'm not sure they want to be highlighted in this manner.
Thanks for the great catch on many typos in the paper.
> I think (but it may be due to my poor english) that the sentence on lines 69-70 misses a "by. in the sense "what can be produced by both ...".
fixed
>On line 106-107 it's written "is now being done" but the references are from 2011 and 2015.
The text is modified. It now reads that this concept was rediscovered with these papers and then I cited recent 2022,2023 papers where it is now being applied.
> On line 182, there's a mention of the plateu phase. Could you explain it better or expand this part?
Great catch. Previously, this was referred to as a long-lived (not plateau) phase. This text is now better tied to the text above.
There's a small typo to correct on line 182 "scneario”
fixed
Reviewer 2 Report
I think that in the introduction, and perhaps in the conclusion (which needs to be expanded anyway), you should mention how our systems for funding grant proposals and publishing research papers create an incentive to go with the bandwagon rather than be a little more iconoclastic. In other words, it is hard to get funding if you are not on the bandwagon as a cheerleader. It is hard to get published if you are not among a small collection of authors who are all publishing similar research.
I do not propose a solution for this, nor do I think you need to, but I do think it is important to mention this as part of an otherwise very interesting and important paper with a strong editorial point to make.
This is a fine paper, but a little editorial scrutiny (for typos, grammer, etc.) is in order.
Author Response
The referee is correct. The funding issue is important. Additional text was added both in the introduction and conclusions to address the funding issue.
The paper was also read over again to catch typos and grammar errors.
Reviewer 3 Report
I understand this interesting manuscript is the author's personal take on the (lack of) development of GRB and SN models. Scientifically it could be improved by adding more specific details in all examples, but the drawback could be that it could distract more from the Story itself. As the Author has provided the relevant references, in principle the details can be obtained by the reader.
Author Response
Initially, I had additional examples, but they diverged from the focused scientific topic.
In the conclusions, I added examples of scientists and scientific work (mostly historical) that were affected by bandwagon fallacies.